# Discrimination of *Schizothorax grahami* (Regan, 1904) Stocks Based on Otolith Morphology

Yang Zhou [1], Li Xu [1], Zhongtang He [1], Weijie Cui [1], Qun Lu [1], Jianguang Qin [2], Shengqi Su [1,3,*] and Tao He [1,3,*]

1   College of Fisheries, Southwest University, Chongqing 400715, China; swu1997zy@163.com (Y.Z.); cqutxuli2020@163.com (L.X.); hzt944581429@163.com (Z.H.); cuiweijie2022@163.com (W.C.); swluqun2023@163.com (Q.L.)
2   College of Science and Engineering, Flinders University, Bedford Park, SA 5001, Australia; jian.qin@flinders.edu.au
3   Key Laboratory of Freshwater Fish Reproduction and Development (Ministry of Education), Key Laboratory of Aquatic Science of Chongqing, Chongqing 400715, China
*   Correspondence: sushengqi@swu.edu.cn (S.S.); hh1985@swu.edu.cn (T.H.)

**Abstract:** Otoliths grow throughout a fish's life and are important for identifying fish stocks and fish age. This study aims to differentiate different stocks of *Schizothorax grahami* (Regan, 1904) in the Chishui River, an upper reach of the Yangtze River, by otolith morphology. The otolith morphology of *S. grahami* from three different river sections was analyzed using the Shape Index, Fourier coefficients, and wavelet coefficients. The composite discrimination success rate of the Shape Index was 59.7%, and it was difficult to distinguish in the scatter plots. In contrast, canonical principal coordinate scatter plots clearly showed three distinguished stocks. The above results indicate that otolith morphology can discriminate between stocks in plateau endemic fish, and several *S. grahami* stocks may be separately managed in the Chishui River.

**Keywords:** *Schizothorax grahami*; otolith; stock discrimination; Fourier coefficient; wavelet coefficient

**Key Contribution:** This study aims to differentiate different stocks of *S. grahami* in the Chishui River; an upper reach of the Yangtze River, by otolith morphology. The otolith morphology of *S. grahami* from three different river sections was analyzed using the Shape Index, Fourier coefficients, and wavelet coefficients. The results indicate that otolith morphology can discriminate between stocks in plateau endemic fish, bridging the research gap in otoliths of *S. grahami*, and several *S. grahami* stocks may be separately managed in the Chishui River.





## 1. Introduction

The otoliths, also known as ear stones, are hard, calcium carbonate structures behind the brains of bony fishes. Otoliths play a role in hearing and balance perception in fish [1,2]. As the fish grows, chemical elements from the environment are continuously deposited on the otoliths through the water and food [3,4]. When the fish is subjected to environmental stress, the otolith morphology is affected by the changes in the physiological conditions of the fish [3–5]. However, during these periods, accumulation problems in otoliths may occur due to possible problematic situations or stress to which the fish is exposed. These accumulation problems (aberrant otolith formation) can cause abnormal otoliths to form in fish and otolith asymmetry. This situation can negatively affect the fish's life, causing prey to escape from them and causing sensory problems. When working on otoliths, the aberration and asymmetry status of the relevant otoliths must be checked [6,7]. Therefore, the morphology of the otoliths and the deposited elements can change with the changing environment.

The earliest method for stock discrimination and classifying fish populations was the body morphology method, which has been gradually replaced by the method of otolith

morphology due to its convenience and accuracy of measurement [8,9]. Moreover, when there are high levels of intraspecific morphological plasticity and low levels of intraspecific genetic differentiation of fish [10,11], then the morphological analysis of otoliths becomes a relatively easy method. For example, *Larimichthys crocea* and *L. polyactis* are morphologically very similar, but the morphology of their otoliths can distinguish them well. *L. crocea* has a needle-like protrusion at the posterior end of the otolith but *L. polyactis* does not [12]. As technology advances, molecular analysis is becoming less expensive and easier to use for species identification, especially when otoliths are difficult to extract from fish (e.g., *Mastacembelus armatus*); the method will make it more straightforward and easier. However, otoliths have a clear advantage when DNA cannot be extracted from fish or when the degree of genetic differentiation within a species is low. Kriwet and Hecht analyzed the morphological characteristics of otoliths preserved in fossils and compared them with existing fish otoliths. They classified the collected fossilized fish species into the Gadiformes (Gadiformes) rat-tailed cod family (Macrouridae), and cross-checked it with the analysis of Broad nasals; their results were consistent [13].

Stocks are a smaller taxonomic category than populations [14]; they are sufficiently numerous to maintain group stability, are capable of self-performing reproduction, individuals within a group share common life history traits [15], and more than one group may exist within a population. Stocks are generally used as a basic unit for fisheries resource management and endangered species conservation. Because otolith morphology has intraspecific differences, it is theoretically possible to use otolith morphometric analysis for stocks delimitation. However, with the development of science and technology and the advancement of morphometrics and image processing techniques, otolith morphology has been successfully used for research at different scales. At present, the mainstream otolith morphology is mainly described using two methods: (i) the traditional Shape Index [16–19], which in general contains less morphological information, is easily affected by the size of the individual, and needs to be corrected using statistical methods; (ii) the wavelet transform and elliptic Fourier transform and other contouring parameters [20–22], which can describe the contouring information, and cannot be affected by the size of the individual, but this method has strong barriers to specialization and is difficult to use. Libungan and Pálsson brought it into use in R in 2015 and developed the corresponding package to make it convenient for describing the morphology of otoliths [23].

The *Schizothorax grahami* (Regan, 1904) belongs to Cyprinidae, Schizothoracinae, and is one of the more primitive species in Cyprinidae [24]. The *S. grahami* is a plateau fish endemic to southeast China and is only found in the upper reaches of the Yangtze River [25]. It commonly lives at the bottom of small rivers with clear water in mountainous areas [26]. Its reproduction and deformation rates are low [27]. The male reaches first sexual maturity at three years old, and the female at four. The first successful artificial breeding occurred in 2007, and a full artificial breeding trial was completed in 2013 [28]. Previous studies on *S. grahami* have focused on growth and development, physiology, biochemistry and disease control. The study of its otolith morphology has not been reported [29]. In this study, the asteriscus otolith of *S. grahami* was selected from the upper, middle, and lower reaches of the source section of the headwater section of the Chishui River. This location (27.42°–27.86°104.77°–105.31° E) belongs to the Yunnan section of the National Nature Reserve for Rare and Endemic Fishes in the upper reaches of the Yangtze River (Northeastern Yunnan Province), with a surface area of 4.82 km$^2$ and an elevation of 751.4–1583.14 m above sea level. The source section of the Chishui River has been a national nature reserve in China since 2005 and has been under continuous resource protection for 17 years. Sampling in this area provides data with minimal anthropogenic disturbance. As *S. grahami* is the dominant species in the basin, its collection will not significantly impact the species diversity or the natural resources.

In this study, we classified the stocks of the caught *S. graham* and compared the otolith morphology of different stocks. This can provide basal data for the conservation of *S. grahami* in the source section of the Chishui River and information for stock identification based on otolith morphology.

## 2. Material and Methods

### 2.1. Study Area and Field Sampling

One hundred and nineteen samples were captured in the headwater section of Chishui River and its tributary (Shikan River and Tongche River) (Figure 1) in March 2022 with gill nets and cast nets. *S. grahami* is the dominant species in the study area, but it is still a key conservation animal in Guizhou Province. For the sustainable development of fishery resources, we cannot overfish and cause damage to local fishery resources. Gender discrimination was not analyzed in this study due to the low number of samples. In this fishing, the minimum total length (TL) of the samples with pearl organs was 102 mm. After measuring the TL (nearest 0.1 mm) and total weight (W, nearest 0.1 g) of each specimen in the field (Table 1), we extracted asteriscus otoliths (left and right) [30] from the inner ear of all specimens with the help of fine forceps in the laboratory. To determine which otolith was used for measurement, a discrimination analysis of the Shape Index was used in both left and right otoliths, and the results showed that left and right otoliths had the same discrimination success rate of 54.9% (Table S1), indicating that there was no significant difference between the two. Thus, we measured the left otolith when available, and the right otolith if the left one was not utilizable or absent. In this study, left otoliths were used for measurement [31,32] After the otoliths were cleaned using an ultrasonic cleaning machine (YM-008S, Motic, Xiamen, China) to remove any additional membranes or surface residues, they were dried and stored in plastic tubes. The samples taken from "a" were treated as the upper reaches stock because there was a drop between "a" and the other sampling sites. The drop was caused by the demolition of the power station; a large amount of rubble was piled up here, which affected the fish migration corridor. Since the DaoLiu River and the TongChe River converge into the mainstream in succession and change the downstream habitats, the samples taken from "I" and "j" were treated as the lower-reaches stock, and the other samples were treated as the middle-reaches stock. All the collection protocols followed the state and institutional guidelines for the care and use of animals and were approved by the Animal Ethics Committee.

**Table 1.** Basic information about the studied samples and studied area.

| Stock | Sampling Sites Number | Number | Total Length Ranges (mm) | Weight (g) |
|---|---|---|---|---|
| Upper reaches (Group I) | a | 34 | 135–212 | 39.8–339.4 |
| Middle reaches (Group II) | b–h | 55 | 82–247 | 11.6–383.6 |
| Lower reaches (Group III) | i–j | 30 | 113–238 | 23.2–328.5 |
| Total | | 119 | | |

Note: The sampling point number in the table corresponds to Figure 1. The samples collected at the upper reaches were Group I, Middle reaches Group II, and Lower reaches Group III.

### 2.2. Otolith Morphometry

The first step in the shape analysis was to capture otolith images using an anatomical microscope (Motic SM171, Motic, Xiamen, China) with a digital camera (Moticam S6, Motic, Xiamen, China). When taking otolith images, the same focal length and magnification were maintained while the lighting conditions were adjusted to produce clearer images. The image was then taken in full color to ensure good focus and high resolution. The otolith must be positioned with the top facing left (Figure 2).

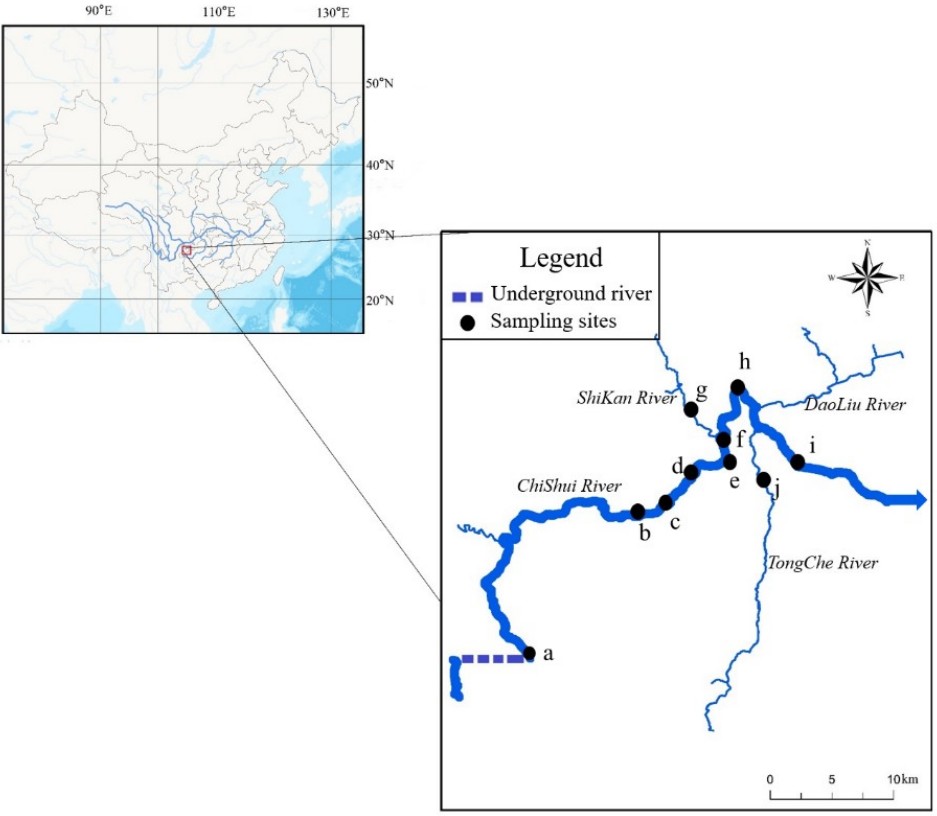

**Figure 1.** *S. grahami* sampling locations (black dots, a–j) in Chishui River. The blue arrow represents the water flow direction.

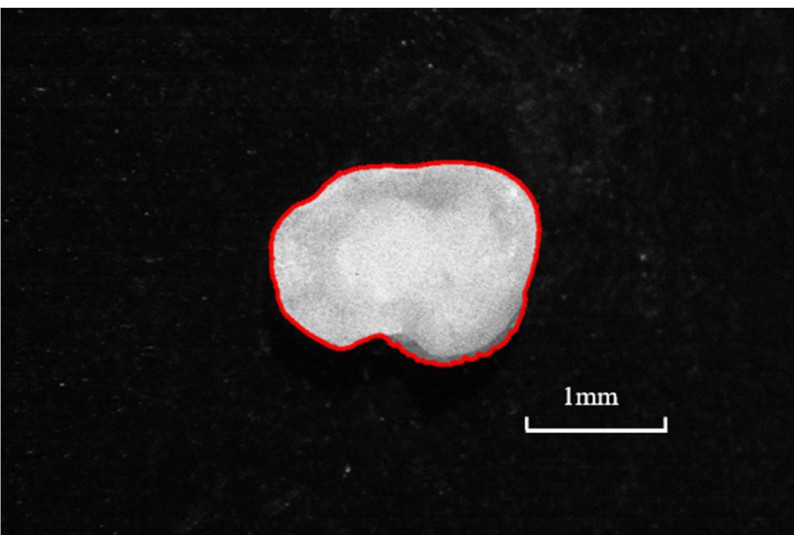

**Figure 2.** Three-dimensional structure of the left otoliths. Red outlines indicate the two-dimensional contour of otoliths.

### 2.2.1. Shape Index

The morphological parameters of the otoliths, such as the otolith area (A), perimeter (P), feret length (FL), feret width (FW), maximum feret length (Fmax), minimum feret length (Fmin), maximum radius (Rmax), and minimum radius (Rmin), were measured by using the micro image analysis software Image-Pro Plus 6.0. The calculation formula of the index is as follows: roundness = $4A/\pi FL^2$, format factor = $4\pi A/P^2$, circularity = $P^2/A$, rectangularity = $A/(FL \times FW)$, ellipticity = $(FL - FW)/(FL + FW)$, radius ratio = Rmax/Rmin,

feret ratio = Fmax/Fmin, and aspect ratio = FL/FW. The measurement method is shown in Figure 3 [17]. By comparing and analyzing the otolith shapes of samples with different body lengths, it was considered that the otolith shapes of each group tended to be stable and could be used for a morphological comparative analysis. The processed image was imported into the shapeR package in Rv4.1.3 [23] after step analysis.

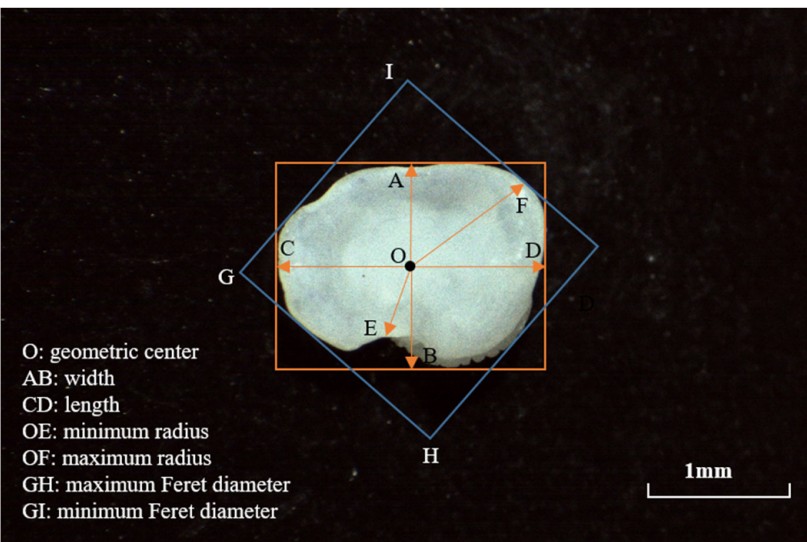

**Figure 3.** Measurement illustration of otolith in *S. grahami*.

### 2.2.2. Elliptic Fourier Coefficients

The shapeR package can provide the first 32 groups of elliptical Fourier harmonics at most, which can satisfy the data analysis requirements. However, if all available Fourier harmonics were selected for subsequent analysis, the number of parameters was too large, so screening was required. This study determined the number of Fourier harmonics for calculating Fourier efficacy. The calculation formula is as follows [33]:

$$P_n = \frac{A_n^2 + B_n^2 + C_n^2 + D_n^2}{2}$$

$P_n$—Efficacy of the NO. $n$ set of Fourier values

$A_n$–$D_n$—Four coefficient of the NO. $n$ set of Fourier values

When the total Fourier efficacy ($\sum P_n$) reached 99.99%, these first $n$ sets of Fourier values were selected. The normalization of the elliptic Fourier coefficients was based on the first three coefficients ($A_1$, $B_1$, $C_1$) of the first set of harmonic values, which was not used in the subsequent analysis. The number of Fourier coefficients used was 4n − 3. In this study, the first set eight of harmonic values was selected.

### 2.2.3. Discrete Wavelet Coefficients

The shapeR package obtained the basic information of the contour by selecting 1024 points on the image contour. According to the pyramid algorithm [34], there were 512 (1024/2 = 512) wavelet coefficients for the highest accuracy, which was the 9th layer ($2^9$ = 512), and one wavelet coefficient for the lowest accuracy, which was the 0th layer ($2^0$ = 1). Therefore, shapeR could give a total of 10 layers of wavelet coefficients. Again, if all the available wavelet coefficients were selected for the subsequent analysis, the number of parameters was too many. In this study, the number of wavelet coefficients used could be determined by calculating the deviation between the otolith contour reconstructed by the wavelet inverse transform and the original image [35]. The deviation was made < 1.0% while controlling the number of parameters, so the first four layers (0–3) of discrete wavelet coefficients were selected uniformly, for a total of 15 coefficients.

*2.3. Statistical Analysis*

Microsoft Excel 2021 was used for data statistics. SPSS 26.0 statistical software was used for one-way ANOVA, principal components analysis, discriminant function analysis, non-parametric tests, and paired two-sample analysis of the mean. Package Vegan in Rv4.1.3 was used for the canonical analysis of principal coordinates. The results were recorded as mean $\pm$ SD. There was no significant difference when $p > 0.05$, $p < 0.05$ is significant, and $p < 0.01$ is very significant.

**3. Results**

*3.1. Analysis of Otolith Morphology Based on the River Section*

3.1.1. Shape Index Analysis

ANOVA

Table S2 shows the results of ANOVA of eight otolith shape indices at a level of significant difference of $p = 0.05$. There was a significant difference in ellipticity, radius, Feret ratio, and aspect ratio, between Group II and Group I, and Group III. There were no significant differences in the other shape indices.

Principal Components Analysis

A principal components analysis of the otolith shape indices showed that two indices had eigenvalues > 1 and were used as the first two principal components, respectively (Table 2). The table shows that the cumulative contribution of the first two principal components was 80.70%, indicating that a small number of indices can be used to summarize the differences in otolith shape in *S. grahami*. In the first principal component, all morphological indicators are greater than in the second principal component, except for the rectangularity of the shape indices, which mainly reflects the degree of difference between the long and short axes of the otolith and the irregularity of the outline.

**Table 2.** Loadings and eigenvalues of the two three-principal components of shape indices of *S. grahami*.

| Otolith Shape Indices | Principal Components | |
|---|---|---|
| | 1 | 2 |
| Roundness | 0.643 | −0.102 |
| Format factor | −0.834 | 0.385 |
| Circularity | 0.827 | −0.409 |
| Rectangularity | −0.467 | 0.610 |
| Ellipticity | 0.895 | 0.433 |
| Radius ratio | 0.936 | 0.061 |
| Feret ratio | 0.926 | 0.190 |
| Aspect ratio | 0.892 | 0.436 |
| Eigenvalue | 5.341 | 1.115 |
| Variance explained | 66.757 | 13.939 |
| Cumulative percentage | 66.757 | 80.696 |

Figure 4 is a scatter plot of the first two principal components, with Group I forming a relatively concentrated group mostly distributed close to the original point. Group II was mainly distributed in the positive direction and was partially intermingled with Group III.

Discriminant Analysis

Using the general discriminant analysis procedure in SPSS, the discriminant equations were derived using the eight otolith shape indicators as independent variables, and the coefficients of the classification functions for the eight groups are shown in Table 3.

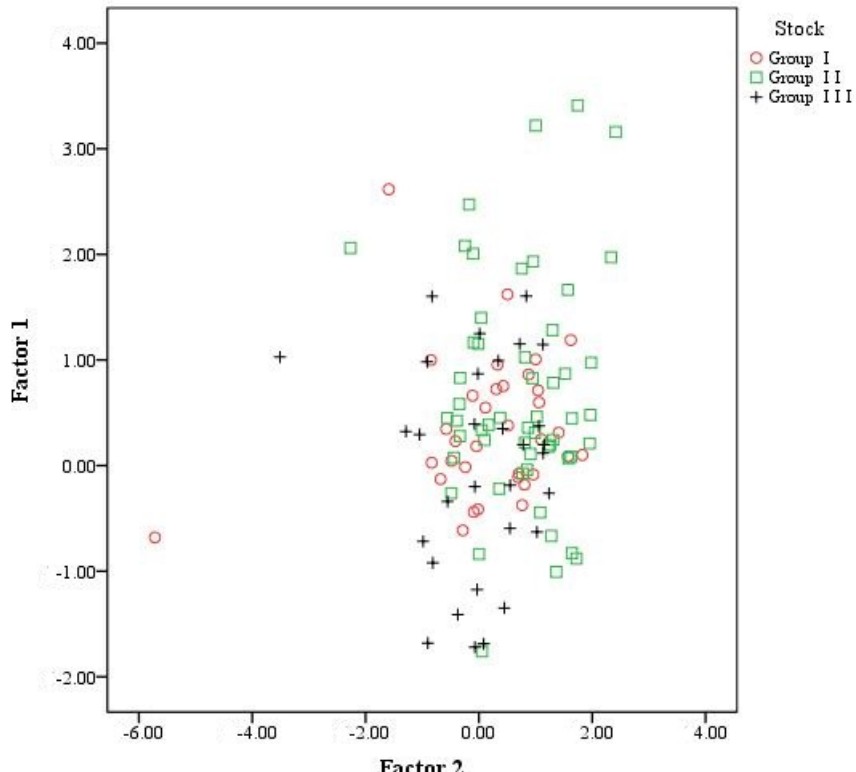

**Figure 4.** Scatter plot of principal components analysis of otolith Shape Index for *S. grahami* with different river sections.

**Table 3.** Parameters of discriminant functions for three stocks of *S. grahami*.

| Shape Index | Stock | | |
|---|---|---|---|
| | Group I | Group II | Group III |
| Roundness | 2.66 | 2.64 | 2.66 |
| Format factor | 37,625.67 | 37,642.49 | 37,640.32 |
| Circularity | 651,841.13 | 652,165.72 | 652,118.05 |
| Rectangularity | 24,936.85 | 24,963.96 | 24,958.10 |
| Ellipticity | 83,462.43 | 83,347.47 | 83,343.42 |
| Radius ratio | 3296.84 | 3308.25 | 3307.26 |
| Feret ratio | 48,436.41 | 48,503.07 | 48,472.72 |
| Aspect ratio | −47,034.44 | −47,014.86 | −47,017.81 |
| Constant | −569,600.38 | −570,250.93 | −570,133.50 |

The otolith shape index of each sample was substituted into the above eight discriminant equations. Eight function values were obtained for each sample, and the group with the largest function value was the group to which the sample belonged. The discriminant results are shown in Table 4. The discriminant rate is 59.7%, except for Group III, which has a slightly lower discriminant rate, but the other groups have a higher discriminant rate.

**Table 4.** Discriminant results for shape indices of *S. grahami*.

| Stock | Number | Accuracy % | Predicted Result | | |
|---|---|---|---|---|---|
| | | | 1 | 2 | 3 |
| Group I | 34 | 61.8 | 21 | 6 | 7 |
| Group II | 55 | 67.3 | 6 | 37 | 12 |
| Group III | 30 | 43.3 | 12 | 5 | 13 |

Figure 5 shows the scatter plot, created according to the first two discriminant function values, from which we can see that the three groups are mixed near the origin. However, Group III is mainly distributed in the positive direction of function 2, Group II is mainly distributed in the positive direction of function 1, and Group I is mainly distributed in the negative direction of the function.

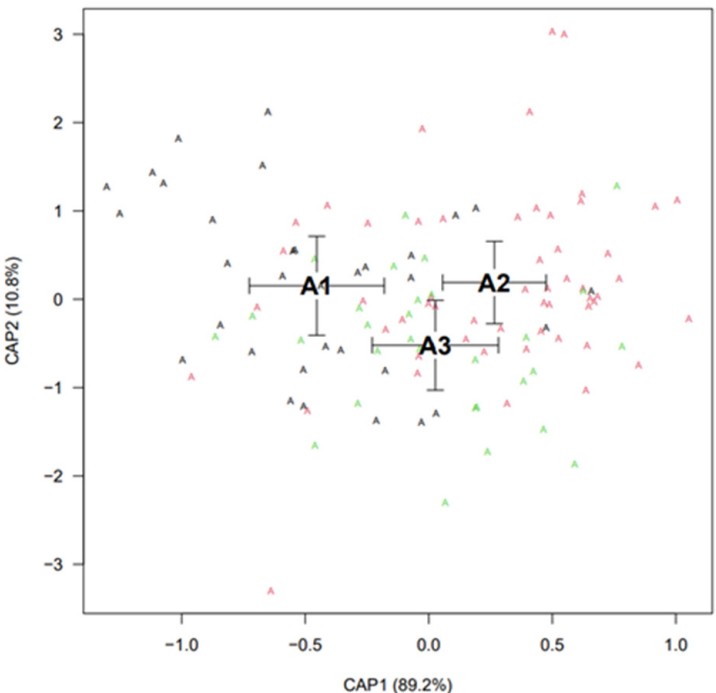

**Figure 5.** Scatter plot of scores based on the first two canonical discriminant functions with different river sections. (The black "A" is "A1", and "A1" is Group I. The red "A" is "A2", and "A2" is Group II. The green "A" is "A3", and "A3" is Group III. The same as below).

### 3.1.2. Fourier Coefficients and Wavelet Coefficients Analysis

ANOVA

An ANOVA was performed on the Fourier coefficients and wavelet coefficients extracted from the three groups (Table S3), and the analysis results showed that the three groups had highly significant differences.

Mean Otolith Shape

The mean shape used in the Wavelet coefficients is plotted in Figure 6a. Otolith contours of the fish from the different studied localities presented differences in mean shape. The morphometry of the otoliths presented modifications in the rostrum-antirostrum (Figure 6b). These modifications were displayed in the wavelet coefficients (ICC) for these regions on the otolith outline at 200°–220° and 310°–340°. The difference between Group II and Group III is smaller at 350°–160°, and the difference between Group I and Group III is smaller at 160°–350°.

Canonical Analysis of Principal Coordinates

Figure 7 displays the canonical analysis of principal coordinates (CAP) scores based on Euclidean dissimilarity indices, including mean scores per location ± standard error. Figure 7a,b show the canonical analysis of principal coordinates for wavelet coefficients and Fourier coefficients, respectively. The first discriminating axis (CAP1) explained 89.2% of the variation among locations, while the second axis (CAP2) explained 10.8% of the variation in Figure 7a. Canonical values for Group I are farthest from the origin in the negative direction of CAP1, Group II in the positive direction of CAP1, and Group III in

the negative direction of CAP2. Similarly, the first discriminating axis (CAP1) explained 73.2% of the variation among locations, while the second axis (CAP2) explained 26.8% of the variation in Figure 7b. Canonical values for Group I are farthest from the origin in the positive direction of CAP1, Group II in the negative direction of CAP1, and Group III in the negative direction of CAP2. In short, the results of the canonical analysis of principal coordinates for both coefficients are similar, and the three groups are partially mixed. However, the analysis of the Fourier coefficients provides a better distinction between the three groups.

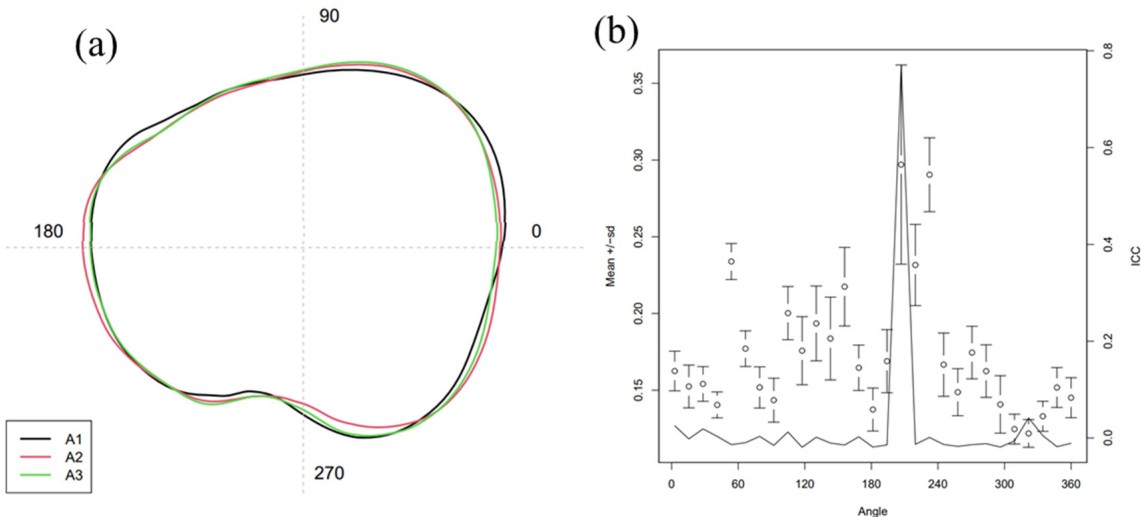

**Figure 6.** (**a**) Black, red, and green contours indicate mean otolith shapes based on wavelet reconstruction for *S. grahami*. (A1 = Group I, A2 = Group II, A3 = Group III) (**b**) Mean and standard deviation (sd) of wavelet coefficients for all combined otoliths and the ratio of between-group variance or intraclass correlation (ICC, black solid line). The horizontal axis represents the angle (°) based on polar coordinates (Figure 6a), where the center of mass of the otoliths is the center point of the polar coordinates.

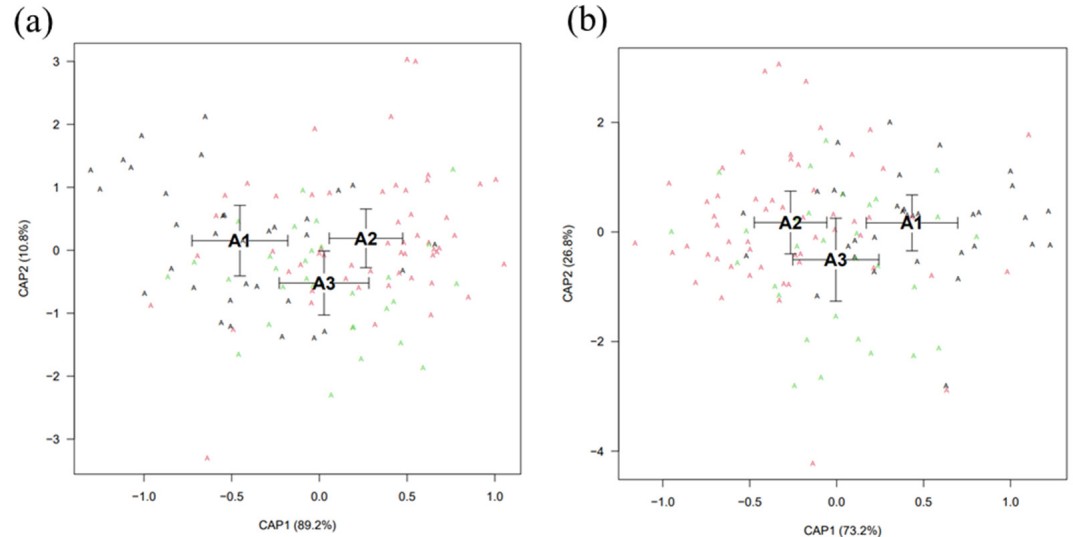

**Figure 7.** Otolith shape of samples from different river sections using canonical analysis of principal coordinates with the wavelet coefficients (**a**) and the Fourier coefficients (**b**). (A1 = Group I, A2 = Group II, A3 = Group III).

## 4. Discussion

The difference in the living environment significantly influenced the shape of the otolith, which was confirmed by the results of this study. The three geographic stocks showed significant differences in the Shape Index. Furthermore, middle reaches showed significant differences in ellipticity, radius ratio, Feret ratio, and aspect ratio from the other two stocks. This indicates that environmental factors influence the degree of variation between the long and short axes of *S. grahami* otolith in this basin [36]. A principal components analysis using the Shape Index method showed that the cumulative contribution of the first two principal components reached 80.70%, indicating that the otolith shape could be better explained. It is similar to the results of the otolith morphology of Chinese and Japanese dotted gizzard shad (*Konosirus punctatus*) [37]. The discriminant analysis with eight shape indices as parameters showed a combined discriminant rate of 59.7%, and the three stocks could be better distinguished. However, some were mixed, indicating morphological similarity, possibly because the three stocks were the same population. Some were not very far away from each other and had similar geographic environments. Chao et al. [38] studied the *Coilia mystus* in the Yangtze Estuary and adjacent waters and showed that the otolith morphology of ChongMing, ZhouShan, and LvSi is highly similar. The discriminant success rate of low reaches was lower at 43.3%, with 12 individuals discriminated as upper reaches. At this point, combined with the ANOVA results of the Shape Index, it indicates that Upper reaches and low reaches have some similarities in terms of otolith morphology.

A more detailed analysis using R was performed to extract the otolith contours of the three different geographical stocks to obtain elliptical Fourier coefficients and discrete wavelet coefficients [23], and their ANOVA results showed highly significant differences between the three stocks. The difference between middle reaches and low reaches is smaller at 350°–160°, and the difference between upper reaches and low reaches is smaller at 160°–350°. From the canonical analysis of principal coordinates with Fourier coefficients and wavelet coefficients, the three stocks are partially mixed, but they can still be separated well from each other. In particular, it is possible to separate upper reaches from low reaches, indicating that the analysis of otolith contours using the Shape Index has some limitations. The Shape Index can provide morphological parameters and describe the otolith contours more vividly. However, the number of commonly used Shape Index parameters is small, and less otolith morphological information can be extracted. Elliptic Fourier transform and discrete wavelet transform can obtain more abundant otolith morphological information and describe more detailed features [12]. Early studies have obtained similar results. For example, in the study of three species of Sciaenids, the discriminant efficiency of the elliptical Fourier coefficient is more than 15.0% higher than that of the Shape Index [12]. However, it increased by 26.6% in the study of Japanese silver croakers [39]. In addition, in some other population studies, the overall discrimination success rate obtained by using elliptic Fourier coefficients exceeded 90.0% [40,41]. Morales et al. conducted research on the otolith morphometric and shape distinction of three redfin species under the genus *Decapterus* (Teleostei: Carangidae); their results showed that the otolith shape analysis was effective in separating the redfin species of *Decapterus* from the Sulu Sea and identifying regions of marked differences in the otolith outline [42]. This research is similar. Munoz-Lechuga found some differences in the otoliths of *Euthynnus* in the Eastern Atlantic and the Mediterranean Sea, based on R language analyses of otoliths [43]. What is more, it has been shown that *Decapterus kurroides* is not the same stock from the Northern Sulu and Southern Sibuyan Seas [44], based on R language analyses of otoliths. All of the above studies have shown that otoliths (R language method) work well for stock discrimination, and that the same population in the same waters may consist of more than one stock.

## 5. Conclusions

In conclusion, otolith morphology can distinguish between stocks of *S.grahami* and find some differences in otolith morphology in different stocks of fish in an area that had

multiple stocks of *S. grahami*. Furthermore, otolith informatics also includes otolith microchemistry, which will be analyzed in conjunction with otolith morphology in the future. We hope the otolith micro-chemistry discrimination of fish stocks is more intuitive and accurate based on otoliths than on otolith morphology.

**Supplementary Materials:** The following supporting information can be downloaded at: https://www.mdpi.com/article/10.3390/fishes8100504/s1, Table S1: Results of discriminant analysis between left and right otoliths; Table S2: ANOVA results of different geographic stocks; Table S3: Comparing otolith shape among three stocks using an ANOVA like permutation test.

**Author Contributions:** Conceptualization: T.H., S.S. and Y.Z. Sampling: Y.Z., Q.L., Z.H. and W.C. Data analysis: Y.Z. and L.X. Preparation of figures and tables: Y.Z. and L.X. Conducting the research, data interpretation, writing: Y.Z., T.H., S.S. and J.Q. Supervision: T.H. and S.S. All authors have read and agreed to the published version of the manuscript.

**Funding:** This research was sponsored by the Biodiversity Survey in the Yunnan Management and Conservation Bureau of National Nature Reserve of Rare and Endemic Fishes in the Upper Reaches of Yangtze River (Yunnan Provincial Department of Agriculture and Rural Development, No. 2023001017), and Natural Science Foundation of Chongqing, China (No. cstc2021jcyj-msxmX0100).

**Institutional Review Board Statement:** The study was approved by the Institutional Animal Care and Use Committe of Southwest University, China (approval code swu-20220225219).

**Data Availability Statement:** The data of the present study are available from the authors upon reasonable request.

**Acknowledgments:** The authors thank Wang Fangyue, Hefei university of technology for the technical support.

**Conflicts of Interest:** The authors declare no conflict of interest. The authors declare that they have no known competing financial interests or personal relationships that could have appeared to influence the work reported in this paper.

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
