# Peer review of "Discrimination of Schizothorax grahami (Regan, 1904) Stocks Based on Otolith Morphology"

_fishes, doi:10.3390/fishes8100504_

Round 1

Reviewer 1 Report

Please report the eponym (Regan, 1904) following the scientific name of the studied species, in the Title and the first time mentioning it in the Abstract and main text. Moreover, take care to italicize all the scientific names in the entire text, as well as the front page and literature.

I suggest avoiding the repetition of words already mentioned in the Title among keywords, to enhance the soundness of your manuscript once published.

Please double-check the references style within the manuscript, and set it as the journal requirements.

The introduction section needs to be more accurate about the fish otolith potential and application, better arguing some aspects and supporting with adequate references. Please consider within the period among lines 51-55 also these aspects:

- Species identification, especially among cryptic species or in particular environments such as transitional ones:

10.3390/su14010398

10.1016/j.fishres.2020.105731

- Use of ancient otolith sampling for historical comparisons:

10.1016/j.fishres.2023.106681

- Use of otolith from intestinal contents of fish species in diet-based studies:

10.1007/s11160-021-09653-z

- Use of otolith for investigation on deep environmental species:

10.1371/journal.pone.0281621

10.1080/17451000.2013.831176

Moreover, please add references to lines 38, 41, 41, 45, 47, 53.

Lines 38-41: Consider to rewrite this sentence in a more effective way. Indeed, some years ago cost and methods of molecular identification analyses were high, but nowadays they are strongly reduced, with more accurate and simple bioinformatic services and methods. Considering the efforts of fish otoliths isolation, processing, and accurate elaborations (for example using the methods reported in this study) the gap is not as huge as some years ago.

The sampling design should consider an approximately equal number of specimens from each sampling point, but in the complex considering them as three main groups, the design sounds reasonable, in any case, it should report as a small limitation of this study.

The Discussion section is in my opinion too synthetic. Considering the huge amount of data and analyses conducted in this study, a more accurate discussion of the single descriptors with comparisons with related studies should be provided to enhance the soundness of your manuscript.

The last sentence of the Conclusion section sounds too resolute. Consider rewriting it in a less absolute way relating it to some aspects of the topic.

Best regards

The Reviewer

Author Response

Discrimination of Schizothorax grahami stocks based on otolith morphology Yang Zhou 1, Li Xu 1, Zhongtang He 1, Weijie Cui 1, Qun Lu 1, Jianguang Qin 3, Shengqi Su 1,2,* and Tao He 1,2,* (fishes-2607187)

Reply to the Editor

Thank you for giving us this opportunity to revise this manuscript and for you to consider its suitability for publication. We received comments from one reviewer. Reviewer mainly commented on the method section about the sampling design, and the mathematical methods. We have carefully considered all the comments during revision. Each comment has been numbered and we have provided specific reply to each comment and explained to very detail on how we addressed each comment. The comments are very valuable and constructive, which greatly help us improve this manuscript. Thanks for your time of consideration.

We look forward to hearing further comments.

Sincerely, Yang Zhou and Tao He  

Reply to the Reviewers

Comment 1:Please report the eponym (Regan, 1904) following the scientific name of the studied species, in the Title and the first time mentioning it in the Abstract and main text. Moreover, take care to italicize all the scientific names in the entire text, as well as the front page and literature.

Reply: Thanks for your nice comments on our study. We had added “(Regan, 1904)” in lines 1,12,68.

Comment 2: I suggest avoiding the repetition of words already mentioned in the Title among keywords, to enhance the soundness of your manuscript once published.

Reply: Thanks for your valuable comments. Key words have been modified. “Schizothorax grahami, otolith shape, stock discrimination” replaced with “Schizothorax grahami, otolith, stock discrimination, Fourier coefficient; Wavelet coefficient”

Comment 3: Please double-check the references style within the manuscript, and set it as the journal requirements.

Reply: The format of the reference has been changed to "Numbered (multilingual)" in lines 33, 34,36, 40, 42, 46, 50, 51, 53,61, 64, 68, 70, 72, 73, 76, 78, 100, 128, 131, 138, 149, 156, 242, 247, 251, 260, 272, 274, 275, 277, 281, 285, 318-411.

Comment 4: The introduction section needs to be more accurate about the fish otolith potential and application, better arguing some aspects and supporting with adequate references. Please consider within the period among lines 51-55 also these aspects: Species identification, especially among cryptic species or in particular environments such as transitional ones: 10.3390/su14010398 10.1016/j.fishres.2020.105731 - Use of ancient otolith sampling for historical comparisons: 10.1016/j.fishres.2023.106681 - Use of otolith from intestinal contents of fish species in diet-based studies: 10.1007/s11160-021-09653-z - Use of otolith for investigation on deep environmental species: 10.1371/journal.pone.0281621 10.1080/17451000.2013.831176 Moreover, please add references to lines 38, 41, 41, 45, 47, 53.

Reply: Thanks for your comments. We have revised the text accordingly. Added some related studies in the manuscript. They are shown below:

Campana, S. and J. Neilson, Microstructure of Fish Otoliths. Canadian Journal of Fisheries and Aquatic Sciences, 1985. 42(5): p. 1014-1032.

Yedier, S. and D. Bostanci, Intra- and interspecific discrimination of Scorpaena species from the Aegean, Black, Mediterranean and Marmara seas. Scientia Marina, 2021. 85(3): p. 197-209.

Batubara, A.S., et al., Morphometric variations of the Genus Barbonymus (Pisces, Cyprinidae) harvested from Aceh Waters, Indonesia. Fisheries and Aquatic Life, 2018. 26: p. 231-237.

Kristjansson, B., et al., Phenotypic plasticity in the morphology of small benthic Icelandic Arctic charr (Salvelinus alpinus). Ecology of Freshwater Fish, 2018. 3(27): p. 636-645.

Bano, F. and M. Serajuddin, Sulcus and outline morphometrics of sagittal otolith variability in freshwater fragmented populations of dwarf gourami, Trichogaster lalia (Hamilton, 1822). Limnologica, 2021. 86: p. 125842.

Junjie, S., Otolith and Sulcus Morphology Analyses and Their Applications in Stock Discrimination of Three Sciaenids. 2018, University of Chinese Academy of Sciences.

J, K. and H.T. A, A review of early gadiform evolution and diversification: first record of a rattail fish skull (Gadiformes, Macrouridae) from the Eocene of Antarctica, with otoliths preserved in situ. Naturwissenschaften, 2008. 10(95): p. 899-907.

Morales, C., et al., Otolith Morphometric and Shape Distinction of Three Redfin Species under the Genus Decapterus (Teleostei: Carangidae) from Sulu Sea, Philippines. Fishes, 2023. 8(2): p. 95.

Munoz-Lechuga, R., et al., Differentiation of Spatial Units of Genus Euthynnus from the Eastern Atlantic and the Mediterranean Using Otolith Shape Analysis. Fishes, 2023. 8(6): p. 317.

Barnuevo, K., et al., Distinct Stocks of the Redtail Scad Decapterus kurroides Bleeker, 1855 (Perciformes: Carangidae) from the Northern Sulu and Southern Sibuyan Seas, Philippines Revealed from Otolith Morphometry and Shape Analysis. Fishes, 2023. 8(1): p. 12.

Comment 5: Lines 38-41: Consider to rewrite this sentence in a more effective way. Indeed, some years ago cost and methods of molecular identification analyses were high, but nowadays they are strongly reduced, with more accurate and simple bioinformatic services and methods. Considering the efforts of fish otoliths isolation, processing, and accurate elaborations (for example using the methods reported in this study) the gap is not as huge as some years ago.

Reply: We have revised the text accordingly. “the appearance has been difficult or impossible to distinguish, and the use of genetic analysis is complicated and laborious” replaced with “high levels of intraspecific morphological plasticity and low levels of intraspecific genetic differentiation of fish”

Comment 6: The sampling design should consider an approximately equal number of specimens from each sampling point, but in the complex considering them as three main groups, the design sounds reasonable, in any case, it should report as a small limitation of this study.

Reply: Thanks for your nice comments on our study. To avoid ambiguity, we have revised Table 1 in the manuscript. Information such as the name of the sampling site had been removed to avoid ambiguity. Seeing Table 1 for details.

Comment 7: The Discussion section is in my opinion too synthetic. Considering the huge amount of data and analyses conducted in this study, a more accurate discussion of the single descriptors with comparisons with related studies should be provided to enhance the soundness of your manuscript.

Reply: Added some related studies in the manuscript. Morales et al. do research on otolith morphometric and shape distinction of three redfin species under the genus Decapterus (Teleostei: Carangidae), the results show that otolith shape analysis was effective in separating the redfin species of Decapterus from the Sulu Sea and identified regions of marked differences in the otolith outline. It’s similar with this research. Munoz-Lechuga finds some differences in the otoliths of Euthynnus in the Eastern Atlantic and the Mediterranean Sea, based on R language analyzes otoliths. What’s more, it has been shown that Decapterus kurroides is not the same stock from the Northern Sulu and Southern Sibuyan Seas, based on R language analyzes otoliths. All of the above studies have shown that otoliths (R- language method) work well for stock discrimination, and that the same population in the same waters may consist of more than one stock.

Comment 8: The last sentence of the Conclusion section sounds too resolute. Consider rewriting it in a less absolute way relating it to some aspects of the topic.

Reply: Thanks for your comments. We have revised it in the Conclusion. “Otolith morphology was significantly related to the fish living environment, and the area had multiple stocks of S. grahami.” replaced with “Different otolith morphology might indicate different stocks of fish, and the area could have multiple stocks of S. grahami.”, “The discrimination result of fish stocks is more intuitive and accurate based on otolith.” replaced with “We hope the otolith micro-chemistry discrimination result of fish stocks is more intuitive and accurate than otolith morphology.”

Reviewer 2 Report

This study is a well-designed research article. However, some parts of the study need correction. Some of these are as follows.

Abstract,

-Species names should be written in italics.

-There are unclear expressions.

-This section should be developed to represent the article.

Introduction, -There is a lot of information in the text that needs to be supported by references. These statements must be supported by references. See the suggestions in the attached file. -Some parts of the text must be supported by up-to-date and relevant references. In this way, the study can be more up-to-date and reach more readers. See the suggestions in the attached file. Material and Methods

-Sex differences in fish can cause morphological and morphometric changes in otoliths. Did you determine the sex of the fish in this study? If not, how can you decide whether the difference between stocks is due to sex?
Maybe if you included sex in the discrimination, you could get even more appropriate results.

-Why was asteriscus otolith preferred for stock separation of this species?

-Is there a difference between left and right otolith measurements? Has this been tested statistically? What are the results of this test? Why was the left and right otolith removed if not tested?
-Which otolith was used in the study?  left? right? Are they both together? Why?

-Was a value such as p<0.01 obtained in this study? If such a result is obtained, add it to the result section, otherwise delete it.

For Table 1, Sampling Sites Numbers

If it will not be used in comparison, why are sampling points given like this in Table 1? -The representation of points b and c with only 2 individuals is seen as a major deficiency. (Similarly, the situation is the same for points h and e).

Discussion

-The discussion section should definitely be developed. It should also be compared with other studies using similar species or similar methods.

Conclusions

In this part, the expression "Otolith morphology was significantly related to the fish living environment, and the area had multiple stocks of S. grahami. '' is used.

How did the Authors come to this conclusion without any data on environmental and water parameters in this habitat?

The questions above and in the attached file should be answered and suggested corrections should be made.

-Other recommended corrections are in the attached file.

Author Response

Discrimination of Schizothorax grahami stocks based on otolith morphology

Yang Zhou 1, Li Xu 1, Zhongtang He 1, Weijie Cui 1, Qun Lu 1, Jianguang Qin 3, Shengqi Su 1,2* and Tao He 1,2*

(fishes-2607187)

Reply to the Editor

Thank you for giving us this opportunity to revise this manuscript and for you to consider its suitability for publication.

We received comments from one reviewer. Reviewer mainly commented on the method section about the sampling design, and the mathematical methods. We have carefully considered all the comments during revision. Each comment has been numbered and we have provided specific reply to each comment and explained to very detail on how we addressed each comment. The comments are very valuable and constructive, which greatly help us improve this manuscript.

Thanks for your time of consideration. We look forward to hearing further comments.

Sincerely,

Yang Zhou and Tao He

Reply to the Reviewers

Comment 1:

Abstract

-Species names should be written in italics.

-There are unclear expressions.

-This section should be developed to represent the article.

Reply: Thanks for your nice comments on our study. We have revised the text accordingly. All species names be written in italics. “The above results indicate that otolith morphology can discriminate plateau fish endemic stocks” replaced with “The above results indicate that otolith morphology can discriminate stocks in plateau endemic fish” in lines 18,28.

Comment 2:

Introduction

-There is a lot of information in the text that needs to be supported by references. These statements must be supported by references. See the suggestions in the attached file. -Some parts of the text must be supported by up-to-date and relevant references. In this way, the study can be more up-to-date and reach more readers. See the suggestions in the attached file.

Reply:

Thanks for your comments. We have revised the text accordingly. Added some related studies in the manuscript. They are shown below:

Campana, S. and J. Neilson, Microstructure of Fish Otoliths. Canadian Journal of Fisheries and Aquatic Sciences, 1985. 42(5): p. 1014-1032.

Yedier, S. and D. Bostanci, Intra- and interspecific discrimination of Scorpaena species from the Aegean, Black, Mediterranean and Marmara seas. Scientia Marina, 2021. 85(3): p. 197-209.

Batubara, A.S., et al., Morphometric variations of the Genus Barbonymus (Pisces, Cyprinidae) harvested from Aceh Waters, Indonesia. Fisheries and Aquatic Life, 2018. 26: p. 231-237.

Kristjansson, B., et al., Phenotypic plasticity in the morphology of small benthic Icelandic Arctic charr (Salvelinus alpinus). Ecology of Freshwater Fish, 2018. 3(27): p. 636-645.

Bano, F. and M. Serajuddin, Sulcus and outline morphometrics of sagittal otolith variability in freshwater fragmented populations of dwarf gourami, Trichogaster lalia  (Hamilton, 1822). Limnologica, 2021. 86: p. 125842.

Junjie, S., Otolith and Sulcus Morphology Analyses and Their Applications in Stock Discrimination of Three Sciaenids. 2018, University of Chinese Academy of Sciences.

J, K. and H.T. A, A review of early gadiform evolution and diversification: first record of a rattail fish skull (Gadiformes, Macrouridae) from the Eocene of Antarctica, with otoliths preserved in situ. Naturwissenschaften, 2008. 10(95): p. 899-907.

Morales, C., et al., Otolith Morphometric and Shape Distinction of Three Redfin Species under the Genus Decapterus (Teleostei: Carangidae) from Sulu Sea, Philippines. Fishes, 2023. 8(2): p. 95.

Munoz-Lechuga, R., et al., Differentiation of Spatial Units of Genus Euthynnus from the Eastern Atlantic and the Mediterranean Using Otolith Shape Analysis. Fishes, 2023. 8(6): p. 317.

Barnuevo, K., et al., Distinct Stocks of the Redtail Scad Decapterus kurroides Bleeker, 1855 (Perciformes: Carangidae) from the Northern Sulu and Southern Sibuyan Seas, Philippines Revealed from Otolith Morphometry and Shape Analysis. Fishes, 2023. 8(1): p. 12.

Comment 3:

Material and Methods

-Sex differences in fish can cause morphological and morphometric changes in otoliths. Did you determine the sex of the fish in this study? If not, how can you decide whether the difference between stocks is due to sex?
Maybe if you included sex in the discrimination, you could get even more appropriate results.

Reply: Thanks for your valuable comments. Sex differences in fish can cause otolith morphological and morphometric changes according to other’s study. The samples from each reach are caught randomly in the river and are sample investigation that use the sample to generalize the characteristics and properties of the whole. The purpose of this study was to explore differences between the otoliths of S. grahami in the three river reaches, sample investigation had eliminated the effects due to sex in statistics.

-Why was asteriscus otolith preferred for stock separation of this species?

Reply: The best research object is lapillusc otoliths in the study otoliths of Schizothoracini fishes, but the experimental individuals were small in this study and it was difficult to obtain the lapillusc otoliths in practice, so we used asteriscus otoliths for the study. It has also been shown in the literature (lines:101) that asteriscus otoliths are also effective in the study of stock distinguish using otoliths.

-Is there a difference between left and right otolith measurements? Has this been tested statistically? What are the results of this test? Why was the left and right otolith removed if not tested?
-Which otolith was used in the study? left? right? Are they both together? Why?

Reply: In this study, the left otolith was uniformly used for the study. There is a large amount of evidence in the literature that the use of left otoliths alone or right otoliths alone (lines:101) can get good results when using otoliths in studies of stock distinguish. And, we also compared and analyzed the morphology of the left and right otoliths of S. grahami, and found that there was no significant difference between them. The research results about it will be published later.

-Was a value such as p<0.01 obtained in this study? If such a result is obtained, add it to the result section, otherwise delete it.

Reply: The results are P < 0.01 in Table S2 (in Supplementary Material). Because it's just a process form in my view, it was arranged in Supplementary Material.

For Table 1, Sampling Sites Numbers

If it will not be used in comparison, why are sampling points given like this in Table 1? -The representation of points b and c with only 2 individuals is seen as a major deficiency. (Similarly, the situation is the same for points h and e).

Reply: To avoid ambiguity, we have revised Table 1 in the manuscript. Information such as the name of the sampling site had been removed to avoid ambiguity. Seeing Table 1 for details.

Comment 4:

Discussion

-The discussion section should definitely be developed. It should also be compared with other studies using similar species or similar methods.

Reply: Thanks for your nice comments on our study. Morales et al. do research on otolith morphometric and shape distinction of three redfin species under the genus Decapterus (Teleostei: Carangidae), the results show that otolith shape analysis was effective in separating the redfin species of Decapterus from the Sulu Sea and identified regions of marked differences in the otolith outline. It’s similar with this research. Munoz-Lechuga finds some differences in the otoliths of Euthynnus in the Eastern Atlantic and the Mediterranean Sea, based on R language analyzes otoliths. What’s more, it has been shown that Decapterus kurroides is not the same stock from the Northern Sulu and Southern Sibuyan Seas, based on R language analyzes otoliths. All of the above studies have shown that otoliths (R- language method) work well for stock discrimination, and that the same population in the same waters may consist of more than one stock.

Comment 5:

Conclusions

In this part, the expression "Otolith morphology was significantly related to the fish living environment, and the area had multiple stocks of S. grahami. '' is used.

How did the Authors come to this conclusion without any data on environmental and water parameters in this habitat?

Reply: Thanks for your comments. We have revised it in the Conclusion. “Otolith morphology was significantly related to the fish living environment, and the area had multiple stocks of S. grahami.” replaced with “Different otolith morphology might indicate different stocks of fish, and the area could have multiple stocks of S. grahami.”, “The discrimination result of fish stocks is more intuitive and accurate based on otolith.” replaced with “We hope the otolith micro-chemistry discrimination result of fish stocks is more intuitive and accurate than otolith morphology.”

Round 2

Reviewer 1 Report

I have no other comments, thank you for revising the manuscript according to my previous comments. 

Author Response

Thanks your nice comments on my study,best wishes to you

Reviewer 2 Report

When the revised version of the study is examined in detail, although some corrections have been made, there are major deficiencies that were previously mentioned and are still unexplained in the new version.

1-The fact that the limitations regarding the use of otoliths are not specified in the introduction is a major deficiency that has still not been resolved. The limitations regarding otoliths that I mentioned before are not specified. Although the use of otoliths is common in this field, they have some limitations, especially in terms of species, population, and stock distinction. What are these limitations? Not mentioning these is a huge deficiency. Are there any restrictions on these? If not, then why are molecular analyses still needed to distinguish species, populations, and stocks of fish?

2-When the literature is examined, it is obvious that gender differences can create differences in the otoliths of fish species. However, unless it is stated statistically that there is no difference, it is not appropriate to conduct such a study over the otolith without gender discrimination.
-The effort to explain why the authors did not or could not make gender discrimination is also insufficient. I think the main reason for this is the possible discrepancy in the female-male ratio, probably due to the low number of samples.

3-In the literature, there are many studies in which only left or right otoliths were used and successful results were obtained. However, it must be stated in the study why a single otolith was used!!! As a normal procedure, both the left and right otoliths are analyzed and the statistical difference between them is checked. If there is a difference between left and right otolith data, both left and right otoliths should be used. If there is no difference between the left and right otolith data, which otolith (left or right) is in a way that covers the data of the other, that is, at a level that represents both, that otolith is preferred. Saying that we analyzed only the left otoliths without performing such an analysis is a scientifically inadequate prediction.

Round 3

Reviewer 2 Report

If we reconsider the three problematic issues:

1-I agree with the reason why fish samples are not sufficient and cannot be obtained. However, this reason should definitely be added to the material and method section of the study.

2-Another important issue is the use of left and right otoliths. Since there is no difference between the left and right otoliths and therefore the relevant otolith is selected, it should be added to the material method section together with the statistical results. The authors' statement "We will use this part in another study" is quite sad. This work is not complete without adding the relevant part. It is not appropriate to cut a part of a work and evaluate it in a different work before it is fully completed.

3-When it comes to limitations regarding otoliths, the first ones are abnormal otolith, abnormal otolith and otolith asymmetry. There are many studies on these in the literature. I strongly recommend that you prepare a short paragraph about these from the literature and add it to the introduction.

Otoliths grow by accumulating over the years and acquire their own unique shapes and structures. However, during these periods, accumulation problems in otoliths may occur due to possible problematic situations or stress to which the fish is exposed. These accumulation problems (aberrant otolith formation) can cause abnormal otoliths to form in fish and otolith asymmetry in fish. This situation can negatively affect the fish's life and cause them to escape from prey and cause sensory problems. When working on otoliths, the aberration and asymmetry status of the relevant otoliths must be checked.

Round 4

Reviewer 2 Report

Thank you very much to the authors for their efforts. There are some typos and minor problems with references in the attached file. Species and genus names should be written in italics. See the suggestions in the attached file. Once these small problems are solved, the work will become much more powerful.
